# Engineering regulatory networks for complex phenotypes in *E. coli*

Rongming Liu[1,4], Liya Liang[1,4], Emily F. Freed [1], Alaksh Choudhury[1], Carrie A. Eckert [1,2] & Ryan T. Gill[1,3 ✉]

Regulatory networks describe the hierarchical relationship between transcription factors, associated proteins, and their target genes. Regulatory networks respond to environmental and genetic perturbations by reprogramming cellular metabolism. Here we design, construct, and map a comprehensive regulatory network library containing 110,120 specific mutations in 82 regulators expected to perturb metabolism. We screen the library for different targeted phenotypes, and identify mutants that confer strong resistance to various inhibitors, and/or enhanced production of target compounds. These improvements are identified in a single round of selection, showing that the regulatory network library is universally applicable and is convenient and effective for engineering targeted phenotypes. The facile construction and mapping of the regulatory network library provides a path for developing a more detailed understanding of global regulation in *E. coli*, with potential for adaptation and use in less-understood organisms, expanding toolkits for future strain engineering, synthetic biology, and broader efforts.

[1] Renewable and Sustainable Energy Institute (RASEI), University of Colorado Boulder, Boulder, Colorado, USA. [2] National Renewable Energy Laboratory (NREL), Golden, Colorado, USA. [3] Novo Nordisk Foundation Center for Biosustainability, Technical University of Denmark, Lyngby, Denmark. [4] These authors contributed equally: Rongming Liu, Liya Liang. ✉email: rtg@biosustain.dtu.dk

Biotechnology applications require engineering complex phenotypes, and thus, a complete reprogramming of innate metabolism. However, the lack of knowledge of the genetic basis of complex traits restricts our ability to modify genes sequentially, and the size of the combinatorial mutational space spanning complex phenotypes is much larger than the size that can be searched on laboratory timescales. Therefore, engineering complex phenotypes at the systems level is a more feasible approach. Without prior knowledge of the targeted phenotype and related genetic basis, directed evolution strategies, such as adaptive laboratory evolution (ALE) and resequencing experiments, have been applied for adapting biological systems toward desired traits[1–3]. However, such strategies only test a small and random fraction of the combinatorial landscape[1]. Engineering of regulator proteins using random mutagenesis and directed evolution methods have further motivated studies for improving complex traits (e.g., global transcription machinery engineering (gTME))[4–6]. Mutations in transcriptional regulators can be harnessed to alter cellular metabolism toward improvement of targeted phenotypes[4–7]. However, these approaches are limited in the number of genes whose expression is affected. Therefore, a strategy is needed to integrate all of the interactions among regulators and compose a comprehensive and sensitive regulatory network to engineer complex phenotypes.

A regulatory network consists of regulators that interact with genes/proteins to control the expression levels of hundreds or thousands of mRNAs and proteins. In *E. coli*, there are >200 regulators with >4000 regulated genes[8,9]. Thus, a highly diverse mutational library targeted to the active sites of different regulators could easily perturb cellular metabolism by altering regulatory networks[10]. Recently, CRISPR-Cas methods have been applied for genome scale, targeted mutagenesis[11–14]. These CRISPR-based methods have high editing efficiency and allow for a diverse set of mutations (e.g., insertions, deletions, point mutations), which enables us to more broadly and deeply investigate the genetic space and advance our understanding of biological systems within the field of synthetic biology and genome engineering.

With increased diversity, the limiting factor for directed evolution applications is how to screen or select for improved variants in a population. If the trait of interest is linked to a fitness increase, then growth selections can be used. Alternatively, screening methods that rely on colorimetric assays or fluorescence-based sorting (FACS) can be employed[15]. Lack of stress tolerance is one of the limiting factors for improving productivity[16,17]. In addition, tolerance can be easily employed for selection studies, whereas directly screening at high-throughput for productivity is often challenging. Therefore, tolerance selections are often used. However, sometimes the identified tolerant strains may not truly be the best ones, and tolerance selections must be followed up with validation of positive variants to determine effects on productivity[4–6].

In this study, we first identify a global regulatory network composed of regulatory genes that regulate or interact with hundreds or thousands of other genes in *E. coli*. We utilize a previously developed CRISPR-based method, CRISPR-Enabled Trackable Genome Engineering (CREATE)[14], to construct a large-scale saturation mutagenesis library targeting the active sites of regulators in the network (Fig. 1a). We then use this regulatory network library to screen and select for different complex traits important for industrial applications (Fig. 1a). We are able to identify both known and previously unidentified mutations that improve tolerance or production phenotypes more than previous methods (Fig. 1a). The resulting global data provide a multidimensional understanding of the *E. coli* regulatory network, uncovers detailed responses to environmental and genetic perturbations, and maps genotype–phenotype linkages in *E. coli* (Fig. 1a). The facile construction and mapping of the regulatory network library provides a path toward the development of a more detailed understanding of global regulation for future strain engineering, synthetic biology, and broader efforts.

## Results

**Regulatory network library design and construction**. *Escherichia coli* is a commonly used model organism. Therefore several databases exist that catalog *E. coli* genetic regulation and operon organization, such as NCBI (https://www.ncbi.nlm.nih.gov/), EcoCyc (http://biocyc.org/), and RegulonDB (http://regulondb.ccg.unam.mx/index.jsp). Using these databases, we chose 82 regulators that regulate or interact with ~4000 genes in total to construct the regulatory network library. We then identified the active sites of these regulators using the above databases, the UniProt (http://www.uniprot.org/), PDB (http://www.rcsb.org/pdb), and Pfam (http://pfam.xfam.org/) databases, and previous studies[18,19] as well as relevant studies in literature that have identified residues or regions of interest using directed evolution approaches. Genes with annotated DNA-binding sites, protein-interface sites, and ligand-binding sites in the NCBI, EcoCyc, and UniProt databases, were input directly into the library design. The genes lacking these annotations were analyzed using the PDB and Pfam databases to predict the potential targeting sites. For genes that had PDB files available, the target sites were predicted around the DNA/ligand-binding regions and dimerization interfaces (<5 Å). For the Pfam database, the target sites were selected using a posterior probability of greater than 90% for the predicted protein domain.

We then used computer-aided design to generate a saturation mutagenesis library targeting the identified regions of the 82 regulators (Supplementary Table 1). This high-resolution library spans a total of 110,120 specific mutations that were predicted to control numerous biological processes, cellular component metabolisms, and molecular functions by perturbing the global regulatory network. By performing selections or screens for a specific phenotype, followed by high-throughput sequencing, all 110,120 specific mutations can be mapped to the desired trait(s). The regulatory network library was constructed in multiplex as described previously (Fig. 1a; "Methods"), and was transformed into *E. coli* MG1655. Overall, the genome editing efficiency of all the sub-libraries was estimated by sequencing both the plasmid barcodes and the genomic sites for 50 randomly picked colonies from each of the four sub-libraries (Supplementary Fig. 1). As a result, the editing efficiency ranged from 30% to 85% and mutational diversity ranged from 30% to 74% in these sub-libraries (Supplementary Fig. 1). Errors in the spacer region of the editing cassettes that arise during oligo synthesis and plasmid construction weaken the gRNA activity and thus decrease the editing efficiency[20] (Supplementary Fig. 2).

**Regulatory network engineering for complex phenotypes**. To demonstrate the utility of the regulatory network library for engineering complex phenotypes, we selected for mutations that confer increased tolerance of furfural, styrene, acetate, isopropanol, and isobutanol. These targeted traits remain important for industrial applications, including overcoming inhibition of cell growth[16], increasing organic solvent tolerance[21], developing low carbon feedstocks[22], and producing biofuel products[23,24]. We performed growth selections with each sub-library under 2 g/L furfural, 300 mg/L styrene, 30 g/L acetate, 30 g/L isopropanol, and 8 g/L isobutanol, all conditions that led to decreased cell growth in wild-type *E. coli* (Supplementary Fig. 3). Plasmids were isolated from the post selection populations, and the barcode-editing

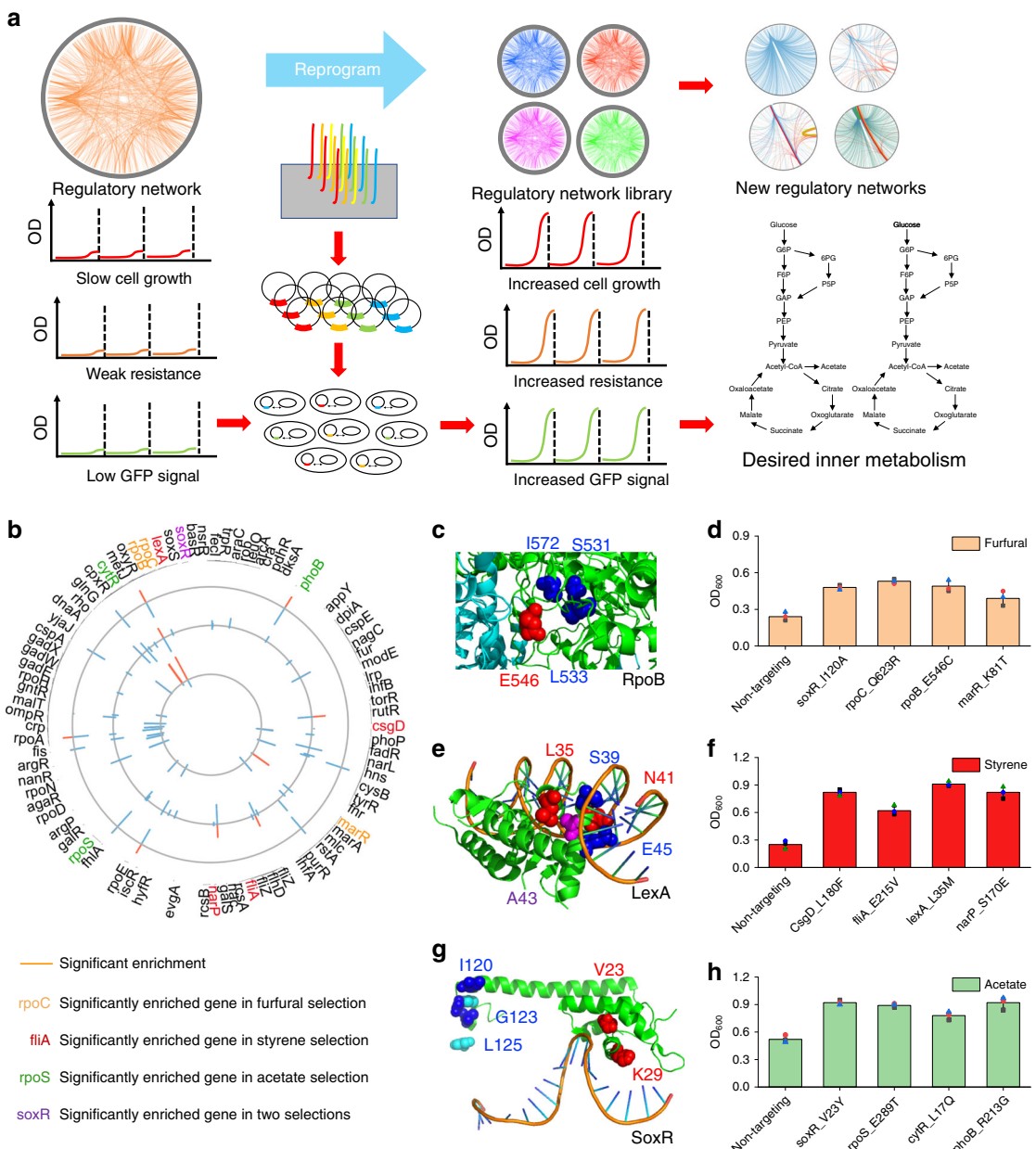

**Fig. 1 Engineering regulatory networks for complex phenotypes in *E. coli*. a** The workflow for engineering regulatory networks for complex phenotypes in *E. coli*. We picked 82 regulators that regulate or interact with ~4000 genes in total to compose a diverse and comprehensive regulatory network library. Then, we designed, constructed, and mapped the regulatory network library containing 110,120 specific mutations expected to perturb metabolism. We screened the library for different targeted phenotypes, and we identified the mutants with desired inner metabolism for the targeted phenotypes. **b** Plasmid barcode-based mapping of enriched variants across all targeted genes under different conditions. From the inner circle to the outer circle, they represent the $\log_2$ enrichment scores of variants under 2 g/L furfural, 300 mg/L styrene, and 30 g/L acetate. The four highly enriched genes for each selection were labeled with different colors. **c** Mutations in RpoB shown on the structural model (PDB: 3IYD). The previously unidentified mutated residue is shown in red, and previously reported residues are shown in blue. **d** Growth verification of the reconstructed variants compared with nontargeting control under 2 g/L furfural. $n = 3$ for each curve. Error bars show mean value ± SD. **e** A partial structural model of LexA (PDB: 3JSO). The previously unidentified mutated residue is shown in red, previously reported residues are shown in blue, and the mutated residue that is enriched in multiple selections is shown in purple. **f** Growth verification of the reconstructed variants compared with nontargeting control under 300 mg/L styrene. $n = 3$ for each curve. Error bars show mean value ± SD. **g** A partial structural model of SoxR (PDB: 2ZHG). The mutated residues for the acetate experiment are shown in red, and the mutated residues for furfural experiment are shown in blue. **h** Growth verification of the reconstructed variants compared with nontargeting control under 30 g/L acetate. $n = 3$ for each curve. Error bars show mean value ± SD.

cassette regions were sequenced to map mutations affecting growth under different stress (Fig. 1b).

Furfural is one of the byproducts from the pretreatment of lignocellulosic biomass, which is highly toxic to microbial hosts[16]. Therefore, the identification of mutations conferring tolerance to these compounds is always a promising target for improving performance of microbial hosts. During the furfural selection, we observed enrichment of multiple mutations in the *soxR*, *rpoC*, *rpoB*, and *marR* genes, all of which have previously been implicated in stress response and antibiotic tolerance[25,26] (Fig. 1b,

c). We found previously unreported mutations located in the E546 residue of the *rpoB* gene under furfural stress, which is in the same region as the I572, L533, and S531 mutations that were identified for enhanced rifampicin resistance[14,27] (Fig. 1c). We then reconstructed the top mutations in MG1655 and tested their furfural resistance. The results showed that the variants grew faster than the nontargeting control under 2 g/L furfural (Fig. 1d).

Styrene is an organic solvent and important monomer used to produce numerous plastics[28]. The global styrene market attained a value of $49 billion in 2018, and is anticipated to rise at a compound annual growth rate of 4.5% during 2019–2025[29]. Although styrene production in engineered *E. coli* has been achieved, the titer and productivity were still not suitable for industrial applications. Thus, construction of a styrene-tolerant strain is important for developing "green" processes for producing styrene[18,28]. Under 300 mg/L styrene, we identified mutations, including in the *lexA* gene, that were consistent with prior reports as well as a range of previously unidentified mutations (Fig. 1b, e)[22]. We reconstructed the positive variants, and the cell growth rate of the variants was above that of the nontargeting control in 300 mg/L styrene that is the inhibitory threshold for *E. coli* (Fig. 1f), which indicated that the styrene production can be further improved without in situ product removal[28,30].

Acetate, in addition to other low carbon compounds such as syngas and methanol, is being examined for use as an alternative feedstock for the production of bioproducts that would improve energy efficiencies[31]. Acetate can serve as a carbon source for *E. coli* production strains; however, the growth rate and biomass yield on acetate are lower than on glucose[22,32]. We analyzed growth selections using 30 g/L acetate as the sole carbon source for increased cell growth (Fig. 1b). We identified positive variants that mapped to the *rpoS*, *cytR*, *soxR*, and *phoB* genes. Some genes have the capability to affect multiple traits, and we showed that mutations in *soxR* were beneficial for both furfural tolerance and acetate utilization, but the location of the mutations was different (Fig. 1b, g). Mutations in the ligand-binding region are enriched under furfural selection, whereas mutations in the DNA-binding region are enriched under acetate selection (Fig. 1g). Next, we reconstructed the enriched mutations in MG1655, and the variants showed increased cell growth rate using acetate as the sole carbon source (Fig. 1h).

Through screening our regulatory network library for targeted traits, we were able to identify numerous mutations that conferred phenotypes, which are important for industrial applications. These results confirmed the universality and convenience of using a regulatory network library. Furthermore, we have identified previously unreported, to the best of our knowledge, genotype–phenotype relationships by altering regulatory networks. If the regulatory network library is used in more applications, we could generate a diverse and comprehensive understanding of the *E. coli* regulatory network at an amino acid level.

**Regulatory network engineering for C3–C4 alcohol tolerance.** Higher-carbon alcohols can be blended with gasoline at a higher volume: ethanol is blended at 10%, whereas isopropanol or isobutanol can be blended at 16%. These higher volumes would lead to reduced greenhouse gas (GHG) emissions given similar biorefinery site and process-specific conditions[33–35]. However, the toxicity of C3–C4 alcohols to *E. coli* is a primary factor limiting titer and productivity for industrial production[36,37]. Thus, it is imperative to discover and utilize genes that confer stress tolerance in order to engineer more effective microbial hosts for biofuel production. Due to the similar structures of isopropanol

and isobutanol, we sought to investigate the similarity between these two alcohols during tolerance selections. We evaluated the selected populations for both enriched genes (genes that had multiple enriched mutations in the top 100 hits) and for individual mutations that were highly enriched. At the gene level, we observed enrichment of multiple mutations in the *rpoE*, *rpoS*, *fis*, and *fur* genes during isopropanol selection and in the *rstA*, *cspA*, *fis*, and *fur* genes during isobutanol selection (Fig. 2a). The *fur* and *fis* genes had a high number of mutations in the top 100 hits in both selections, with L122 in the *fur* gene (Fig. 2b) as well as K25 and L55 in the *fis* gene (Fig. 2c) being in the top ten most enriched mutations (from their subpools) in both selections. At the individual mutation level, we also found a similar trend for the mutation enrichment in the C3–C4 selection, with some mutations, such as RpoE_D182Y and FliA_R94K, being highly enriched in both selections (Fig. 2d, e).

To confirm our population-level analyses, we reconstructed the ten variants containing mutations with the highest enrichment or mutations from the gene with highest hits in both selections. We then tested growth in 30 g/L isopropanol and 8 g/L isobutanol (Fig. 2f), and all of the reconstructed variants had growth rates at or above those of the nontargeting control (Fig. 2f). Among them, the variants FliA_R94K and RpoA_G87W grew >50% faster than the nontargeting control under isopropanol and isobutanol stress (Fig. 2f). *fliA* and *rpoA* are RNA polymerase sigma factors, which are initiation factors that promote the attachment of RNA polymerase (RNAP) to specific initiation sites and are then released[38,39]. These results indicated that these mutations likely changed the transcription of other genes thereby causing the enhancement of isopropanol and isobutanol tolerance in *E. coli* strains, as will be discussed more below. Since enhanced tolerance does not always lead to enhanced production, we introduced these mutations into MG1655 strains with either an integrated isopropanol (PA14) or isobutanol (IB500C) production pathway (Supplementary Table 2). We then performed flask fermentation studies on each variant. We determined that the mutant FliA_R94K produced isopropanol and isobutanol titers (g/L) ~50% higher than that of the nontargeting control (Supplementary Fig. 4).

**FliA mutations improve C3–C4 alcohol tolerance.** In *E. coli*, *fliA* encodes a flagellum-specific sigma factor that controls the expression of flagellum-related genes[40]. Other high-level regulators such as CRP, H-NS, OmpR, and CsgD are also involved in the regulation of flagellar synthesis and of *fliA* (Supplementary Fig. 5). Two mutations in the *fliA* gene were highly enriched in the isopropanol and isobutanol-tolerance selections (Fig. 2e), which indicated that the related regulatory network was reprogrammed. Thus, we used global transcriptional analysis to investigate the metabolism of alcohol resistance in the FliA_R94K mutant.

Under alcohol stress, there were >1000 genes up-/down-regulated more than twofold in the FliA_R94K mutant compared with the nontargeting control (Fig. 3a), which indicated that the FliA_R94K mutation perturbed global metabolism. Thus, we first focus on the regulatory network for FliA (Supplementary Fig. 5). The *fliA* gene was significantly upregulated (>100-fold) in the mutant, with the transcription level increasing with an increase in alcohol stress (Fig. 3b). In the regulatory network for FliA, the expression of the *fliA* gene is directly regulated by FlhDC, NsrR, and CsgD[40,41] (Supplementary Fig. 6). NsrR and CsgD repress the expression of the *fliA* gene[41], while upregulation of the *flhDC* genes causes upregulation of the *fliA* gene[40]. In the FliA_R94K mutant, *flhD* and *flhC* are both upregulated, which is likely responsible for the observed increase in *fliA* (Supplementary

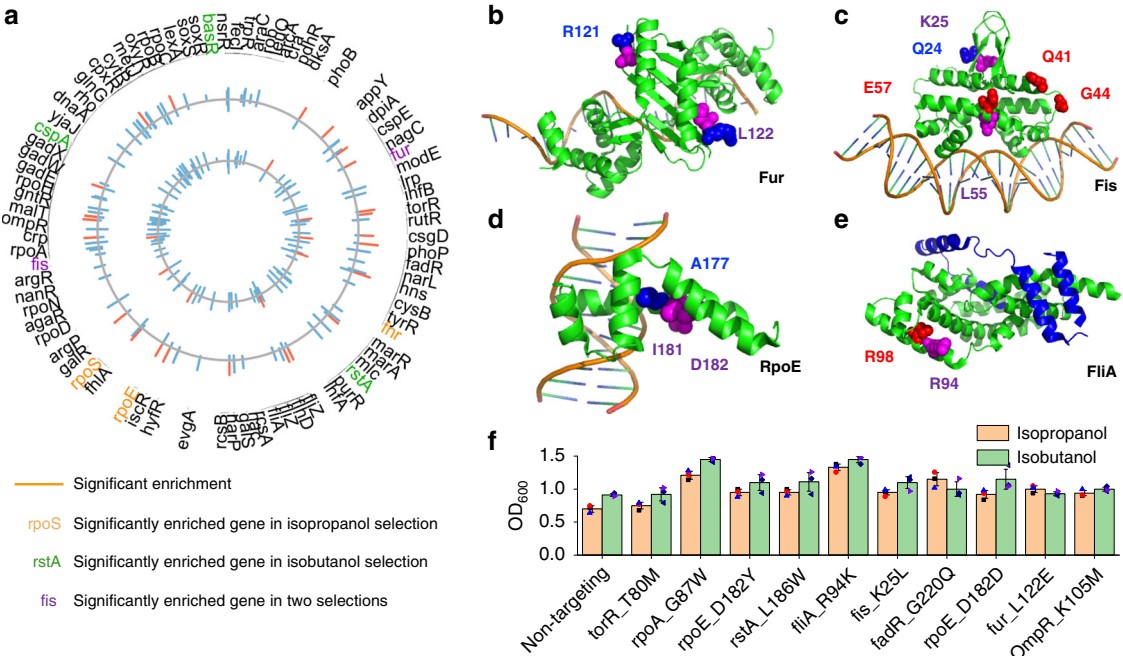

**Fig. 2 Mapping the effect of regulator library genes for enhanced tolerance to isopropanol and isobutanol. a** Plasmid barcode-based mapping of enriched variants across all targeted genes under different conditions. The plots represent the log₂ enrichment scores of variants under 30 g/L isopropanol (inner circle) and 8 g/L isobutanol (outer circle). **b–e** The partial structural models of **b** Fur (PDB: 4RB1), **c** Fis (PDB: 5E3O), **d** RpoE (PDB: 2H27), and **e** FliA (PDB: 1RP3). The mutated residues for isopropanol experiments are shown in red, the mutated residues for isobutanol experiments are shown in blue, and residues that are enriched in both experiments are shown in purple. **f** Growth verification of the reconstructed variants compared with nontargeting control under 30 g/L isopropanol or 8 g/L isobutanol. $n = 3$ for each curve. Error bars show mean value ± SD.

Fig. 6). In addition, higher-level regulators in the network, CRP and RpoS, as well as other global regulators, Fnr, Lrp, and Fur, were all upregulated in the FliA_R94K mutant, which could be the main reason for thousands of genes with altered transcriptional levels (Supplementary Fig. 6).

Alcohol-induced stress significantly inhibits the cell growth and alcohol production of biofuel-producing bacteria. Generally, alcohol damages cell wall and membrane integrity, decreases cross-membrane proton gradients, weakens cell respiration and ATP generation, and alters synthesis of macromolecules[42–45]. Thus, we further investigated the genes with altered expression in the mutant FliA_R94K (Fig. 3c–e). For the genes associated with protein and membrane metabolic processes, there were >150 genes with significant transcriptional changes in the mutant. Furthermore, there were 75 genes related to cell wall structure, membrane composition, and membrane transport that were upregulated in the mutant, and which could improve cell stability under alcohol stress (Fig. 3c). For the genes associated with energy metabolic processes, there were >200 genes up-/down-regulated more than twofold in the mutant. Among them, six of the seven ATP synthesis genes and 49 genes related to cell respiration were upregulated in the mutant (Fig. 3d). Moreover, the ATP concentration was increased ~threefold compared with the nontargeting control after 24 h under 30 g/L isopropanol and 8 g/L isobutanol (Supplementary Fig. 7), which further confirmed that the FliA_R94K mutant had higher ATP biosynthesis capability under alcohol stress.

In addition to the above genes, many genes previously identified as being involved in the stress response had large transcriptional changes in the mutant under alcohol stress. Upregulation of nfo[46], ung[47], dam[48], yrfG[49], and dinG[50], which are related to DNA replication and repair, as well as bcp[51], grxC[52], marR[53], and marA[53], which are related to oxidative stress response and multidrug resistance, might improve the resistance

to the alcohol stress (Fig. 3e). In addition, we also compared the global transcription of FliA_R94K and the nontargeting control with/without isobutanol stress (Supplementary Fig. 8). We observed that the majority of genes with altered transcriptional levels in the mutant in the presence of isobutanol were not significantly changed under the no-stress condition, which indicated that this mutation causes alcohol-inducible changes in transcription, and helped cell survival under alcohol stress (Supplementary Fig. 8). Interestingly, 21 genes related to tRNA synthesis were all upregulated in the mutant without isobutanol, which could improve protein synthesis under no-stress conditions (Supplementary Fig. 9).

**Regulatory network engineering for isobutanol production.** After the isopropanol and isobutanol-tolerance selections, we sought to use the regulatory network library for directed screening/selection of isopropanol and isobutanol production. The use of in vivo biosensors to drive the engineering of microbial cell factories was successfully applied to the discovery of alcohol production[54]. Unfortunately, the alcohol-dependent-growth biosensor (pSelect-1) is not sensitive to isopropanol production. Thus, we used the alcohol biosensor to select for increased isobutanol production by coupling isobutanol-induced TetA and GFP expression to E. coli growth rate. The E. coli isobutanol-production strain, IB500C ΔadhE, harboring the pSelect-1 plasmid (containing a tetA-gfp cassette under the control of an alcohol-regulated promoter $P_{BMO}$) displayed isobutanol-dependent changes in growth phenotype and GFP expression (Supplementary Fig. 10).

We performed selections for isobutanol production after transforming the regulatory network library into IB500C ΔadhE. During the selection in 96-well plates, we determined that the library variants produced more alcohol than the nontargeting

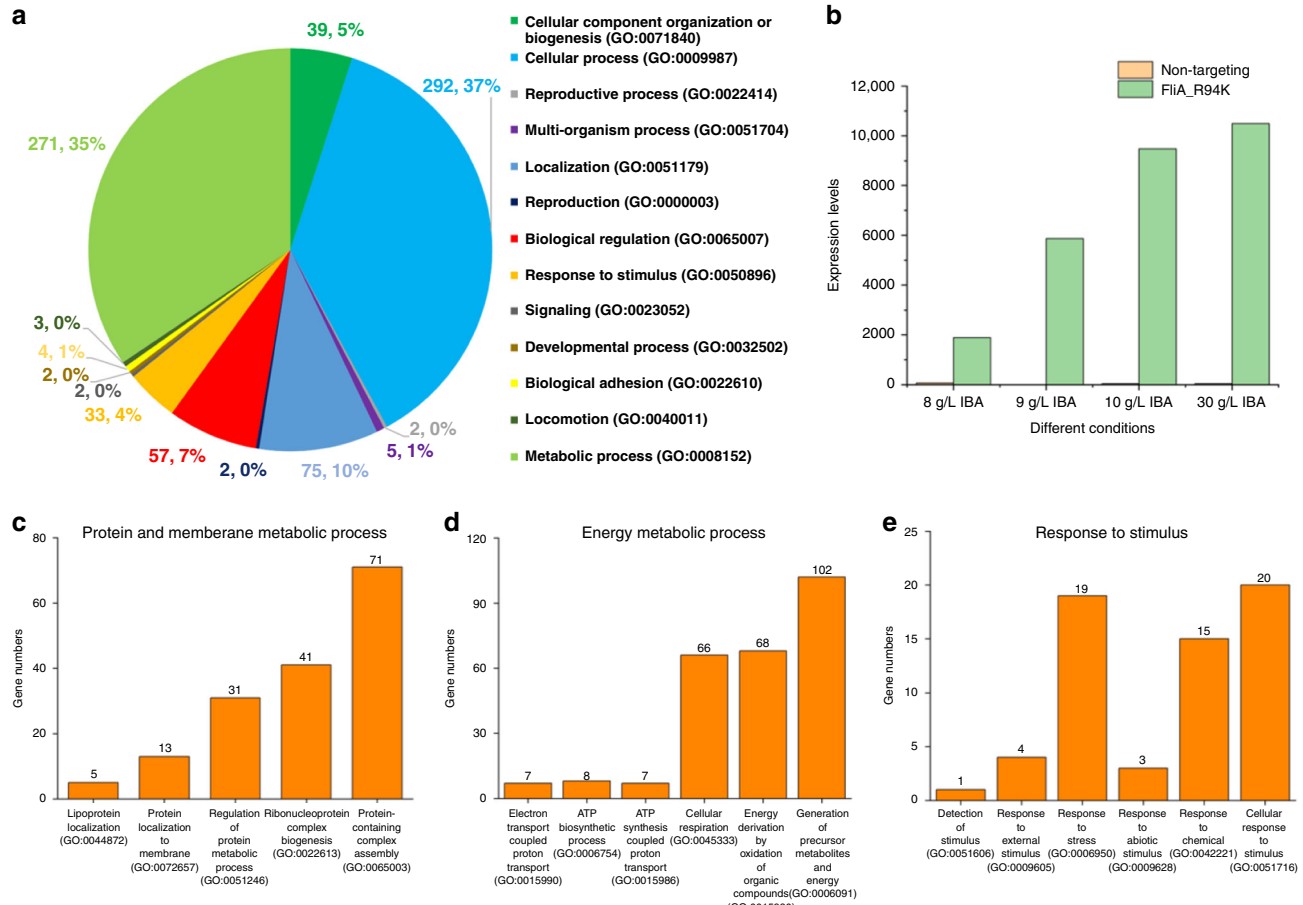

**Fig. 3 Transcriptional analysis of the FliA_R94K mutant that led to improved C3–C4 alcohol tolerance. a** Pie chart showing categories of genes that were up-/downregulated more than twofold in the FliA_R94K mutant compared with the nontargeting control. The numbers from the pie chart are gene numbers and the percentage of each category for genes with significant transcriptional changes. **b** The number of transcripts for the *fliA* gene in the FliA_R94K mutant and nontargeting control under different conditions. **c–e** Number of genes with significant transcriptional changes for different biological processes: **c** protein and membrane metabolic process, **d** energy metabolic process, and **e** response to stimulus.

control due to the higher ratio of GFP/OD$_{600}$ in the wells of library variants (Supplementary Fig. 11). Plasmids were isolated from the post selection populations, and the barcode-editing cassette regions were sequenced to map the mutations affecting isobutanol production (Fig. 4a). We observed that mutations in the *soxR*, *cpxR*, *glnG*, *ompR*, *mlc*, and *rstA* genes were the most enriched in the selections (Fig. 4a). All of these genes are transcription factors, and most of them are different from the genes selected under high isobutanol stress. Mutations in the *rstA* gene were enriched in both the isobutanol-tolerance and production selections (Fig. 4a). We then reconstructed the strain IB500C Δ*adhE* with the enriched 12 variants from the production selection, and tested them for isobutanol production. The results showed that all of the reconstructed variants produced isobutanol above the level of the nontargeting control (Fig. 4b). Moreover, the variants OmpR_S181K produced 10.6 g/L isobutanol, which was ~1.4-fold higher isobutanol compared with the nontargeting control (Fig. 4c). This is the largest improvement for isobutanol productivity without modifying the enzymes in the central carbon metabolism and isobutanol pathway reported to date[55–57].

**RstA mutations improve isobutanol tolerance and production**. In both the isobutanol-tolerance and production selections, mutations in the *rstA* gene were enriched (Fig. 4a). We measured enrichment scores of mutations by amino acid position in RstA

(Fig. 4d, e; Supplementary Fig. 12). We expected to observe one of two general patterns: enrichment for (i) high-fitness substitutions of a small number of residues in both selections, suggesting the same mutations confer benefits for both tolerance and production; or (ii) substitution of residues in different regions, suggesting that specific mutations affect either isobutanol tolerance or production, respectively. We observed the latter in our data, which suggested that the location of mutations could affect different phenotypes during the selection of isobutanol tolerance and production (Fig. 4d, e; Supplementary Fig. 12). In addition, we used PROVEAN (Protein Variation Effect Analyzer)[58] to evaluate enriched mutations in the *rstA* gene in both the isobutanol-tolerance and production selections. Of the enriched mutations, ~62% of mutations in the isobutanol-tolerance selection and ~69% of mutations in the isobutanol-production selection are predicted to be deleterious mutations, which could be loss-of-function mutations. The remainder of the mutations are predicted to be neutral mutations, which might not affect gene function, or could potentially be gain-of-function mutations (Supplementary Fig. 13).

RstA is part of a two-component system, a signal transduction system that bacteria use to adapt to external stimuli (Supplementary Fig. 14). Each two-component system is composed of a sensor protein-histidine kinase (HK) and a response regulator (RR), which form a signal transduction pathway together via a histidyl–aspartyl phospho-relay[59]. RstA is an RR and RstB is its

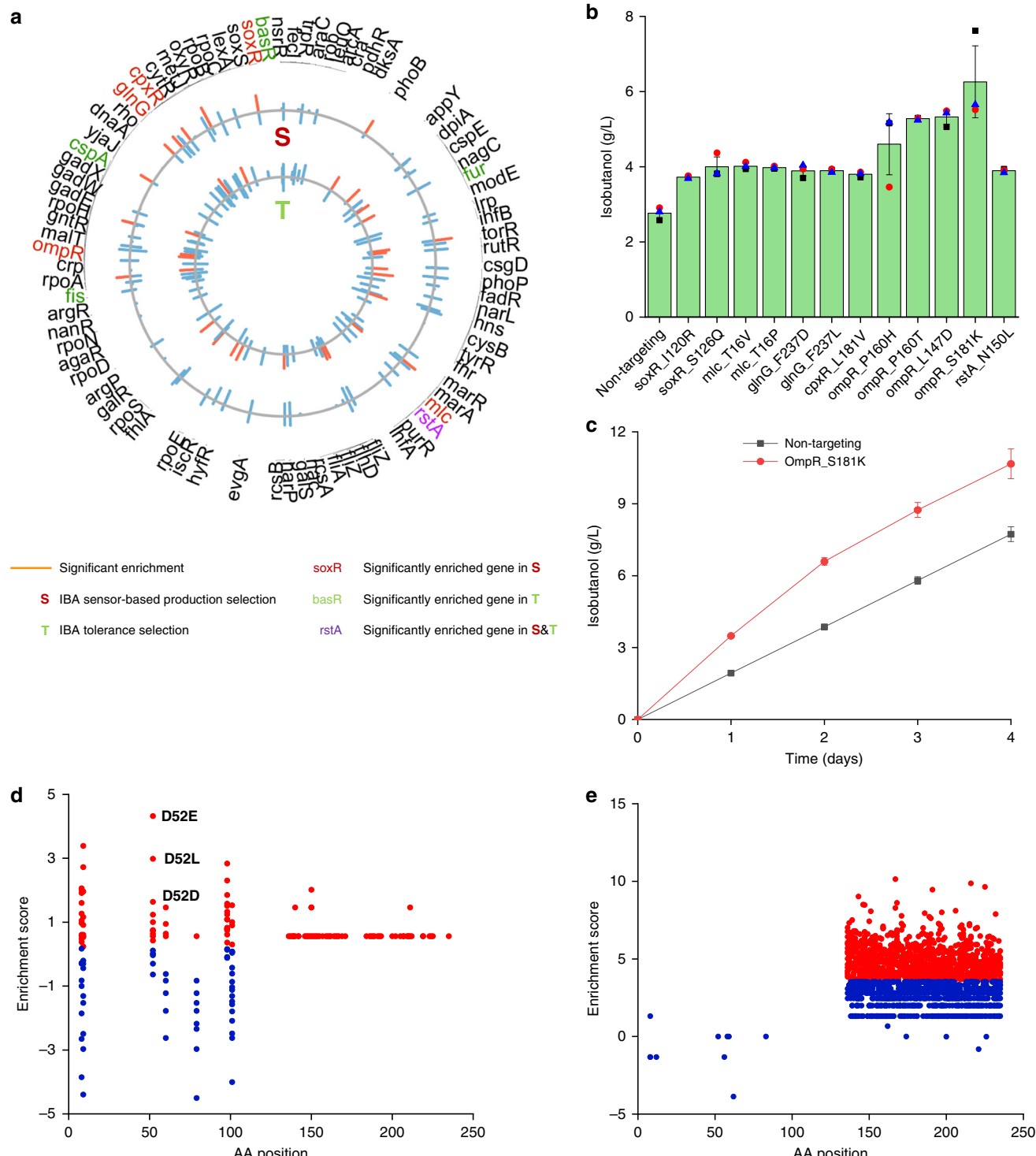

**Fig. 4 Mapping the effect of regulator library genes for enhanced isobutanol tolerance and production. a** Plasmid barcode-based mapping of enriched variants across all targeted genes under different conditions. The plots represent the log$_2$ enrichment scores of variants with enhanced isobutanol tolerance (inner circle) and production (outer circle). **b** Isobutanol fermentation using different variants after 48 h (three biological replicates). Error bars show mean value ± SD. **c** Fed-batch fermentation of OmpR_S181K and nontargeting control for isobutanol production. $n = 3$ for each curve. Error bars show mean value ± SD. **d, e** Mapping of enrichment across positions that were targeted in the *rstA* gene for (**d**) the isobutanol-production selection and (**e**) the isobutanol-tolerance selection. We used a weighted enrichment score to separate the mutations in red (>weighted enrichment score) and blue (<weighted enrichment score) ("Methods").

cognate HK; together, they function as a membrane sensor[59,60]. RstA consists of an N-terminal receiver domain (RD) and a C-terminal DNA-binding domain (DBD). Based on the predicted structure, the mutations conferring increased isobutanol production were mainly located in the RD (Fig. 4d), whereas the mutations conferring enhanced isobutanol tolerance mainly were located in the DBD (Fig. 4e).

Phosphorylation of RstA-P$^{Asp52}$ in the RD induces dimerization, which allows two RstA DBDs to bind to the RstA box, and then regulate the expression of downstream genes. The most enriched mutations in the isobutanol-production selection, such as D52E and D52L, likely affect the phosphorylation of RstA (Fig. 4d; Supplementary Fig. 12a). Furthermore, D52D (GAC-> GAT) is another highly enriched mutation (Fig. 4d). Although this synonymous mutation does not change the amino acid residue, other studies have reported that synonymous codon substitutions can perturb protein folding and impair cell fitness[61,62]. In addition, mutations in E8 and Y98 were also highly enriched during the selection. Based on the predicted structure of RstA, these mutations are close to D52 and therefore possibly affect the phosphorylation reaction.

In *E. coli*, RstA can be induced by low extracellular levels of Mg$^{2+}$ via the PhoQ/PhoP two-component regulatory system, and then upregulate acid shock genes and the biofilm regulator gene, *csgD*, under acidic conditions[63–65]. The top 50 enriched mutations from the isobutanol-tolerance selection were located in the DBD (Y136-L232) of RstA based on the predicted structure (PDB: 4NHJ) (Fig. 4e; Supplementary Fig. 12b). We tested one of these mutations, RstA_L186W and found that it upregulated the transcription of *csgD* (Supplementary Fig. 15), which could help cells survive under stress conditions[64].

**OmpR mutations improve isobutanol production.** OmpR is a member of the two-component regulatory system EnvZ/OmpR involved in osmoregulation[66]. OmpR plays a central role in both acid and osmotic stress responses[67,68], which activate the transcription of *ompF* at low osmolarity and *ompC* at high osmolarity[67,69]. The mutants OmpR_P160T, OmpR_L147D, and OmpR_S181K produced more isobutanol compared with other reconstructed variants and the nontargeting control, which suggested that there could be a universal mechanism in the *ompR* variants that led to enhanced isobutanol production. Thus, we analyzed global changes in gene expression for all three variants (Fig. 5a).

There are 550 genes up-/downregulated more than twofold during the fermentation process. We classified the genes into different groups based on the Gene Ontology (GO) biological process (Fig. 5a). Many altered transcripts were found in genes related to cellular component organization, biological regulation, response to stimulus, and signaling (Fig. 5a). Contrary to the tolerant variants, the OmpR mutants had few genes significantly change expression in response to isobutanol stress, but had a large change in transcription in the absence of isobutanol (during fermentation) (Fig. 5b, c). Furthermore, genes related to stress resistance, i.e., those with GO terms such as response to stimulus, DNA replication, regulation of transcription, and protein modification process, were downregulated in all three variants during fermentation (Fig. 5a). These responses are opposite to the transcription changes in the FliA_R94K mutant, where upregulated stress-resistance genes might improve the alcohol tolerance (Fig. 3e). This could be the main reason why these three mutations were not enriched in the tolerance selection.

In the isobutanol-producing pathway, pyruvate is the key substrate from central carbon metabolism. Thus, we investigated the effect of mutations on pyruvate metabolism. Generally, the

genes in glycolysis are upregulated (Fig. 5a), which could increase the metabolic flux from glucose to pyruvate (Fig. 5d). Furthermore, the accumulation of pyruvate is dependent on the balance between its biosynthesis and consumption. For all three mutants, *pykA*, which converts phosphoenolpyruvate to pyruvate, was upregulated, as was *pdhR*, which represses the reaction from pyruvate to acetyl-CoA (Fig. 5d). These transcriptional changes suggested that there was more metabolic flux for conversion of pyruvate to isobutanol. In addition, the biological process of aerobic respiration was upregulated in the mutants, which could allow the strain to gain more NADH for isobutanol production (Fig. 5a). In addition, the upregulated aerobic respiration and ATP synthesis genes could further improve the energy supply for cell growth and metabolism, which could also benefit isobutanol production (Fig. 5a).

The isobutanol-production pathway also utilizes the 2-keto acid precursors of native amino acids; thus amino acid biosynthesis also affects alcohol production. The downregulation of arginine, glutamate, and glutamine biosynthetic processes (Fig. 5a) can restore the accumulation of α-ketoglutarate (Fig. 5d) in the TCA cycle and further improve ATP and NADH synthesis derived from aerobic respiration. Similarly, the downregulation of phenylalanine and serine biosynthetic processes (Fig. 5a) can increase the metabolic flux in glycolysis and the PPP, and further improve the substrate and cofactor pools for isobutanol production (Fig. 5d).

## Discussion

Living cells are the product of gene expression programs involving regulated transcription of thousands of genes. The hierarchical relationship between transcription factors, associated proteins, and their target genes can be described as a regulatory network. Numerous studies have determined how most of the regulators encoded in *E. coli* associate with genes across the genome in living cells[40,70–73]. When the regulatory network is perturbed, cellular metabolism will be reprogrammed. Studying this network allows us to understand how a genotype and the environment are integrated to regulate downstream physiological responses. Previous efforts have focused on up-/downregulation of regulators or mutating a handful of important regulators for specific phenotypes[40,70–73]. However, there were no universal tools that could be used to engineer diverse traits. In this study, we designed and constructed a regulatory network library consisting of a total of 110,120 specific mutations in 82 regulators. There are ~4000 genes that interact in this regulatory network. We then used this library for different targeted phenotypes that are important for industrial applications, and identified a series of positive mutations in a single round of selection which improved these traits, including overcoming inhibition of cell growth (furfural), increasing organic solvent tolerance (styrene), developing low carbon feedstocks (acetate), and improving biofuel resistance and production (isopropanol and isobutanol).

Therefore, we have shown that the use of a regulatory network library can be universally applicable and is convenient and effective for engineering targeted phenotypes. The regulatory network library will also allow for further study of genotype–phenotype relationships and could lead to significant gains in understanding the mechanisms of a wide variety of cellular responses. Due to the complexity of the regulatory network library, we analyzed the data at both the gene and amino acid levels as a way to determine genotype–phenotype relationships. At the gene level, the enriched regulators in the experiments showed relevance in their functions, regulations, or interactions (Fig. 6a). For example, the highly enriched genes in furfural experiments interacted with each other (Fig. 6a), and all

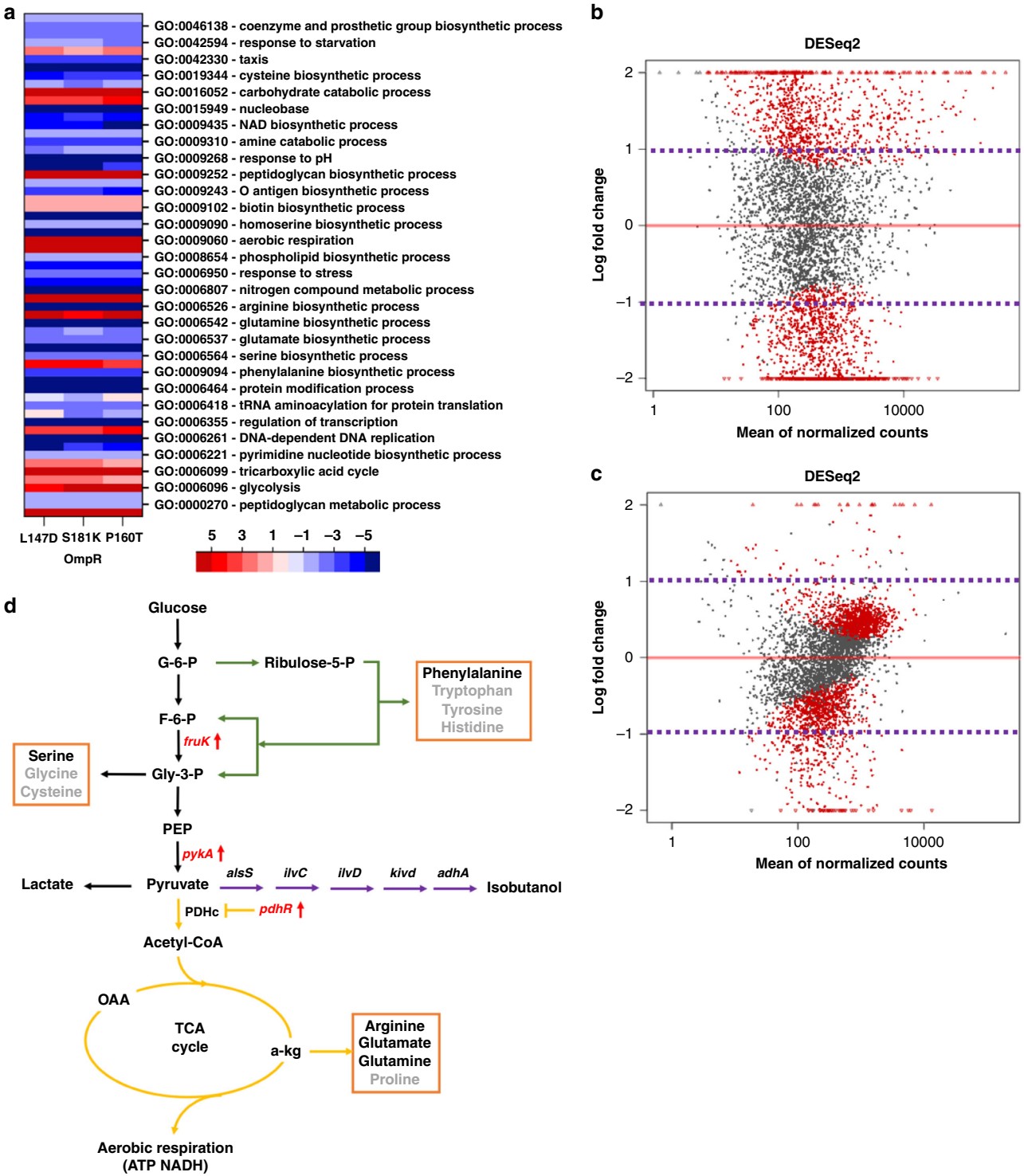

**Fig. 5 Transcriptional analysis of the OmpR mutants that led to improved isobutanol production. a** Heatmap showing genes categories whose transcription changed greater than twofold (enrichment > 1 and < −1) in all three variants compared with the nontargeting control. **b, c** The DEseq2 plots for the OmpR_L147D mutant (**b**) without and (**c**) with isobutanol stress. The purple line indicates the cutoff for significantly enriched transcripts ($E_i$ > 1 or < −1). **d** The metabolic pathway for isobutanol production in engineered *E. coli*.

contributed to stress and antibiotic tolerance[25,26]. Furthermore, we observed that some regulators showed the capability of multistress resistance in the different conditions (Fig. 6b).

At the amino acid level, there were two behaviors for the enriched mutations: (i) the mutations were in the same region for the same or similar targeted traits, or (ii) the mutations were located in different regions for the different targeted traits. For the first

situation, we observed mutations in the same region of Fur, Fis, and FliA, in both isobutanol and isopropanol-tolerance experiments, and in the isobutanol-production experiment, we saw mutations in the same region in OmpR (Supplementary Fig. 16). For the second scenario, we observed that the *soxR* gene was enriched in the furfural and acetate experiments, and the *rstA* gene was enriched in the isobutanol-tolerance and production experiments, but the

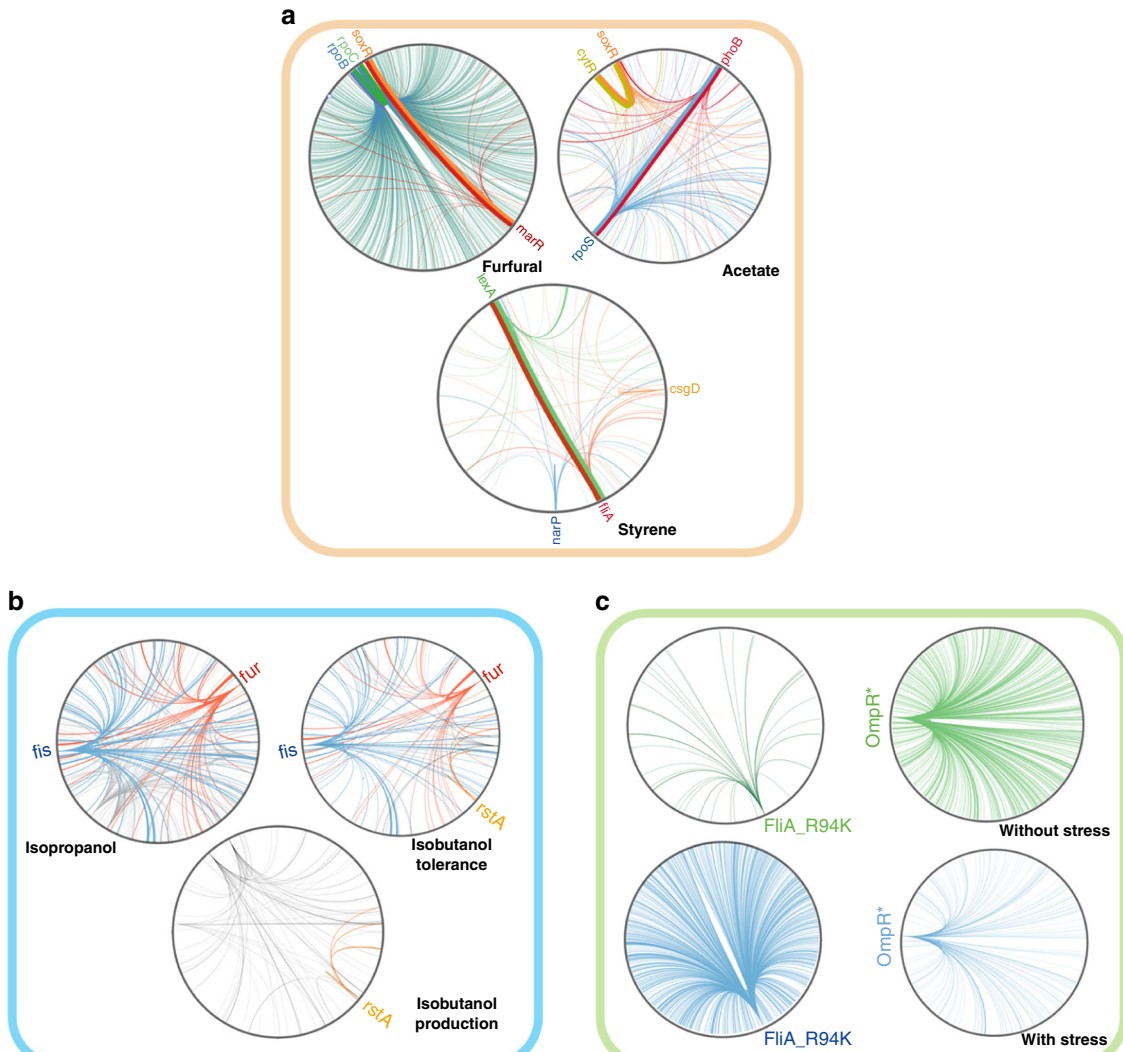

**Fig. 6 Behaviors of the regulatory network library for complex phenotypes.** Each circle represents the *E. coli* regulatory network. For clarity, only the regulators that were highly enriched in each screen/selection are labeled on the outside of the circle. The lines show interactions between the labeled regulators and genes in the regulatory network (as determined from databases or literature) under different conditions, with the interactions labeled in the same color as the enriched regulator. Interactions between regulators are shown with bold lines. **a** Some enriched regulators interacted with other top enriched regulators, suggesting that these regulators may have similar functions in conferring resistance to stress. **b** Some enriched regulators conferred multistress resistance at the gene level in the different conditions. The *fis* and *fur* genes are enriched in both the isopropanol and isobutanol-tolerance selections. The *rstA* gene is enriched in both the isobutanol-tolerance and -production selections. **c** The function of some enriched mutations showed high correlation to the environment. The transcription change in the mutant FliA_R94K is strong under isobutanol stress, but weak without isobutanol stress. In contrast, the transcription change in the ompR mutants, such as the mutant OmpR_S181K (shown here as a representative mutation), is weak under isobutanol stress, but strong without isobutanol stress.

mutations were in different regions for each selection (Fig. 1g; Supplementary Fig. 12). Thus, we could further sub-classify the regulatory network library into different groups of genes or mutations, depending on the targeted traits (Fig. 6). In addition, the behaviors of enriched mutations were likely related to the environment (Fig. 6c). Without isobutanol stress, the OmpR mutations affected the regulatory network to reprogram central metabolism, whereas these mutations hardly perturbed the regulatory network under isobutanol stress (Fig. 6c). Thus, the variants OmpR_S181K produced 10.6 g/L isobutanol, which was ~1.4-fold higher isobutanol compared with the nontargeting control (Fig. 4c). In contrast, FliA_R94K appears to be alcohol-induced, and only modifies transcription under alcohol stress (Fig. 6c).

The results of this study suggest that certain regulatory genes, such as *fliA* and *soxR*, play an important role in conferring

tolerance to multiple different growth inhibitors. Therefore, they may be good targets for more directed mutagenesis for other tolerance phenotypes for future strain engineering efforts. Furthermore, the mutations identified in this study could potentially be combined with other known mutations to further tolerance to, and/or production of, a desired compound. Since the CREATE strategy used here can be performed iteratively[19], the regulator library could also be used to make combinatorial mutations, which could potentially lead to greater improvements in the desired phenotype. Multiple applications of this library could provide a multidimensional understanding of the *E. coli* regulatory network, and uncover detailed responses to environmental and genetic perturbations. The regulatory network library allows for rapid adaptation to new conditions and genotype–phenotype mapping of *E. coli* strains.

Finally, the facile construction and mapping of the regulatory network library provides a path for developing a detailed understanding of global regulation in less-understood organisms for future synthetic biology and broader efforts. The mutations identified here could be adapted for use in other microbes as part of strain engineering efforts. In addition, a regulatory network library similar to the one described here could potentially be adapted for any microbe, in which efficient CRISPR-Cas mediated genome editing is possible. Many industrially relevant bacteria have recently had CRISPR-Cas systems developed, including *Clostridium thermocellum*[74], *Streptomyces* species[75], *Corynebacterium glutamicum*[76], *Zymomonas mobilis*[77], and *Pseudomonas putida*[78]. Since the regulatory network library can be applied to improve any phenotype that can be effectively selected or screened for, it opens the door for efficient strain engineering for complex phenotypes in *E. coli* and other microbes.

## Methods

**Library design and computationally identified information**. The saturation mutagenesis library sites (Supplementary Table 1) were selected using the NCBI (https://www.ncbi.nlm.nih.gov/), EcoCyc (http://biocyc.org/), UniProt (http://www.uniprot.org/), PDB (http://www.rcsb.org/pdb), and Pfam databases (http://pfam.xfam.org/), and previous studies[18,19] as well as relevant studies in literature that have identified residues or regions of interest using directed evolution approaches. For the NCBI, EcoCyc, and UniProt databases, the labeled DNA-binding sites, protein-interface sites, and ligand-binding sites were input into the library design. The genes without labeled sites were analyzed using the PDB and Pfam databases to predict the potential targeting sites. For genes that had PDB files available, the target sites were predicted around the DNA/ligand-binding regions and dimerization interfaces (<5 Å). For the Pfam database, the target sites were selected using a definitive posterior probability (higher than 90%) in predicted protein domain.

**Library plasmid construction**. Oligo pricing was available at ~$0.10–0.15/oligo for pooled oligo synthesis products (available from Agilent Biotechnologies, Santa Clara, CA, USA). Each oligo pool was amplified with a subpool oligonucleotide (e.g., BC1_F) and Insert_R (Supplementary Table 3). The reaction conditions were 98 °C for 60 s; ten cycles of 98 °C for 30 s, 60 °C for 30 s and 72 °C for 90 s; ten cycles of 98 °C for 30 s and 72 °C for 90 s; and 72 °C for 5 min. Unincorporated oligos and ssDNA were subsequently removed from the libraries using a QIAquick PCR cleanup kit (Qiagen, Valencia, CA, USA).

The linearized backbone was amplified using the bbF and bbR primers (Supplementary Table 3). The reaction conditions were 98 °C for 30 s; 30 cycles of 98 °C for 10 s, 60 °C for 30 s and 72 °C for 90 s; and 72 °C for 2 min. This backbone was treated with DpnI and purified using the QIAquick Gel Extraction Kit (Qiagen, Valencia, CA, USA).

The circular polymerase extension cloning (CPEC) reaction was used for assembly. The reaction conditions were 98 °C for 30 s; ten cycles of 98 °C for 10 s, 60 °C for 30 s and 72 °C for 90 s; and 72 °C for 2 min. The reactions were desalted using dialysis by spotting the reaction on a 0.025-μm pore filter floating in ddH₂O. Following desalting, the cloned products were electroporated into *E. cloni* 10 G ELITE electrocompetent cells (Lucigen Corporation, Middleton, WI, USA). Libraries were plated onto LB with 100 μg/mL carbenicillin to estimate transformation efficiency. The library plasmids were purified using a QIAprep Spin Miniprep Kit (Qiagen, Valencia, CA, USA). All PCR steps were performed with the high-fidelity Phusion enzyme (New England Biolabs, Ipswich, MA, USA) to ensure production of a high-quality library.

**Library construction and selection**. The host *E. coli* MG1655 strain carried the pSIM5 (Supplementary Table 2) and pX2-Cas9 plasmids (Supplementary Table 2). When OD₆₀₀ reached 0.5–0.6, expression of the pSIM5 plasmid was induced by shaking the cells for 15 min at 42 °C. After chilling on ice for 15–30 min, the cells were washed twice with 20% of the initial culture volume of ice cold ddH₂O. Then, the library plasmids (or single editing plasmid) were mixed with the cells, followed by chilling on ice for 5 min. Following electroporation, the cells were recovered in SOB medium for 2 h. Then, 1 μL of cells was plated to determine transformation and editing efficiencies, and the remaining cells were transferred into a 10× volume of LB with 50 μg/mL kanamycin and 100 μg/mL carbenicillin. The overnight cultures were centrifuged and resuspended in fresh LB or other media for target selection. Before selection, the genome editing efficiency of the constructed library was tested by sequencing 50 colonies in each subpool.

Following an overnight recovery, the cells were harvested by centrifugation and resuspended in a fresh selection medium. All selections were performed in conical tubes (CELLTREAT Part no. 229475) or flasks and were inoculated at an initial OD₆₀₀ of 0.1. Two serial dilutions (over 48–96 h, depending on the growth rate in the target condition) were performed for each selection, with OD₆₀₀ = 0.1 as the

initial inoculum concentration. The selections were performed in LB or M9 medium. The LB or M9 medium here was supplemented with 30 μg/mL kanamycin and 100 μg/mL carbenicillin, plus the chemical for the selection. The cells were harvested by pelleting 1 mL of the final culture, and the cell pellet was boiled in 100 μL of TE buffer to preserve both plasmid and genomic DNA for further analyses.

**Library sequencing and data analysis**. Custom Illumina compatible primers were designed to allow a single amplification step from the editing plasmid with assignment of experimental reads using barcodes (Supplementary Table 4). The editing cassettes were amplified directly from the plasmid sequences of boiled cell lysates using the following conditions: 98 °C for 30 s; 20 cycles of 98 °C for 10 s, 60 °C for 30 s and 72 °C for 90 s; and 72 °C for 2 min. Amplified fragments were verified by 1% agarose gel electrophoresis, cleaned using a QIAquick PCR Cleanup Kit and processed for NGS using standard Illumina preparation kits. Illumina sequencing and sample preparation were performed with the primers listed in Supplementary Table 4.

Paired-end Illumina sequencing reads were sorted according to the Golay barcode index with an allowance of up to three mismatches and then merged using the USEARCH fastq_merge algorithm (http://www.drive5.com/usearch/). Sorted reads were then matched against the database of designed editing cassettes using the usearch_global algorithm at an identity threshold of 90%, allowing up to 60 possible hits for each read. The resulting hits were further sorted according to percent identity, and read assignment was made using the best matching editing cassette design at a final cutoff of 99.5% identity to the initial design. It should be noted that this read assignment strategy attempts only to identify correlations among the designed genotypes and may therefore miss other important features that arise due to mutations during the experimental procedure. This approach was chosen both to simplify data analysis and to evaluate the "forward" design and annotation procedure and its ability to accurately identify meaningful genetic phenomena. Alternative variant-calling algorithms may enable further investigation into the underlying genetic diversity in future applications.

For each individual biological replicate, enrichment scores were calculated as the logarithm (base 2) of the ratio of the frequency of post selection to preselection. Frequencies were determined by dividing the read counts for each variant by the total experimental counts.

The enrichment score ($E_j$) was calculated as follows:

$$E_j = \frac{\log_2(Y_j)}{\log_2(X_j)},$$

where $X_j$ is the frequency of plasmid j before selection in the deep-sequencing measurement, and $Y_j$ is the frequency of plasmid j after selection in the deep-sequencing measurement.

A weighted average was used to combine the enrichment scores obtained in the two biological replicates, according to the following formula:

$$Wavg = \frac{\sum_{i=1}^{N} ci \times wi}{\sum_{i=1}^{N} Ci},$$

where $Wavg$ is the weighted average enrichment score, $i$ is the biological replicate, $C$ is the read count obtained for the variant in the biological replicate, and $W$ is the enrichment score calculated for the variant in the biological replicate.

To assess significance, the average enrichment scores for all synonymous mutations included in the library were calculated (average μ of wild-type enrichment). Bootstrap analysis (resampled with replacement 20,000 times) was performed to obtain a 95% confidence interval for the wild-type enrichment average μ. Variants were considered significantly enriched if their weighted enrichment scores were at least μ ± 2*σ (i.e., P-value ≤ 0.05 assuming a normal distribution of synonymous mutation enrichment scores), with σ being the standard deviation.

**RNAseq and real-time qPCR**. RNAseq: RNA was isolated using an RNeasy Mini Kit (Qiagen, Valencia, CA, USA), then diluted to 60 ng/μL in a total volume of 15 μL RNase-free water, and 1 μL of SUPERase-IN (Life Technologies) was added. The RNAtag-Seq protocol was followed for tagging, pooling, ribosomal depletion, and library construction using the RNA oligo linkers described in the supplementary information[79]. The Ribo-Zero treatment was performed using the Ribo-Zero rRNA Removal Kit for Gram-negative bacteria (Illumina). All oligos were purchased from IDT. The final, pooled sample was sequenced on a single Illumina MiSeq V3 150-cycle run with a 5% PhiX spike-in. RNA-seq data were analyzed and normalized using Rockhopper[80].

qPCR: The total RNA was isolated using an RNeasy Mini Kit (Qiagen, Valencia, CA, USA). RNA preparations were checked for DNA contamination via PCR with primers targeting the 16S rRNA gene. The cDNA synthesis was then performed using a SuperScript II cDNA synthesis kit (New England Biolabs, Ipswich, MA, USA) with 200 ng of the total RNA. The qPCR was performed in triplicates using a 7500 fast real-time PCR system (Applied Biosystems, Foster City, CA) with the primers listed in Supplementary 2; the expression levels of targeted genes were normalized to the 16S rRNA gene levels of their respective samples.

**Mutant reconstruction**. The cassettes (Supplementary Table 2) were ordered as separate gBlocks from IDT. Each cassette was transformed into the target host strain and then subjected to the same growth conditions as indicated above for the pooled library selection.

**Alcohol fermentation procedures**. For isopropanol production, shake flask experiments were carried out in a rotary shaker at 220 rpm and 37 °C using 250-mL conical flasks, each containing 25 mL of SD-8 medium. SD-8 medium (NH$_4$Cl, 7.0 g/L; KH$_2$PO$_4$, 7.5 g/L; Na$_2$HPO$_4$, 7.5 g/L; K$_2$SO$_4$, 0.85 g/L; MgSO$_4$·7H$_2$O, 0.17 g/L; trace elements, 0.8 ml/L; yeast extract; 10 g/L) containing 2% glucose was used for fermentations. The trace element solution contained the following (in grams per liter of 5 M HCl): FeSO$_4$·7H$_2$O, 40.0; MnSO$_4$·H$_2$O, 10.0; Al$_2$(SO$_4$)$_3$, 28.3; CoCl$_2$·6H$_2$O, 4.0; ZnSO$_4$·7H$_2$O, 2.0; Na$_2$MoO$_4$·2H$_2$O, 2.0; CuCl$_2$ · 2H$_2$O, 1.0; and H$_3$BO$_4$, 0.5. For antibiotic selection, the concentrations of antibiotics were 100 µg/mL carbenicillin, 34 µg/mL chloramphenicol, and 30 µg/mL kanamycin. The pH was maintained at ~6.0 throughout the fermentation by the addition of 50% NaOH solution when required.

Isobutanol production from strains was measured after 24 or 48 h growth at 37 °C in M9 medium (glucose, 80 g/L; NaCl, 0.5 g/L; Na$_2$HPO$_4$·12H$_2$O, 17.1 g/L; KH$_2$PO$_4$, 3 g/L; NH$_4$Cl, 2 g/L; MgSO$_4$·7H$_2$O, 246 mg/L; CaCl$_2$·2H$_2$O, 14.7 mg/L; FeSO$_4$·7H$_2$O, 2.78 mg/L supplemented with 10 g/L yeast extract. The growth was performed in 50-mL conical tubes containing 25 mL of media. Fed-batch fermentations were initiated as a batch culture with an initial glucose concentration of 20 g/L in 250-mL flasks containing 50 mL of M9 media supplemented with 10 g/L yeast extract. The temperature was maintained at 30 °C, and the pH was maintained at ~7.0 by the controlled addition of a 50% NaOH solution. The glucose solution was fed into the flask when the residual glucose concentration decreased to 0–2 g/L, and the final concentration was ~20 g/L.

**Analytical methods**. Glucose and alcohol were analyzed by HPLC (LC-20 AD with a refractive index detector RID-10A, Shimadzu, Kyoto, Japan) with a Bio-Rad Aminex HPX-87H column at 65 °C. The mobile phase was 5 mM H$_2$SO$_4$ at a flow rate of 0.6 mL/min. All the samples were centrifuged at $15,800 \times g$ for 6 min, and then filtered through a 0.22-µm filter before analysis.

**Reporting summary**. Further information on research design is available in the Nature Research Reporting Summary linked to this article.

## Data availability

The source data underlying Figs. 1, 2, 4 and Supplementary Figs. 3, 4, 7, 15 are provided as a Source Data file. High-throughput sequencing data have been deposited in the Sequence Read Archive (SRA) under accession codes: SAMN15447118, SAMN15447119, SAMN15447120, SAMN15447121, SAMN15447122, SAMN15447123, SAMN15447124, SAMN15447125, SAMN15447126, SAMN15447127, SAMN15447128, SAMN15447129. Data supporting the findings of this paper are available from the corresponding authors upon reasonable request. Source data are provided with this paper.

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

## Acknowledgements

This project was sponsored by the US Department of Energy (Grant DE-SC0018368).

## Author contributions

R.L., L.L., and R.T.G. developed the concept. R.L., L.L., A.C., and C.A.E. all aided in the design of experiments. The experiments were done by R.L., L.L., and E.F.F. The data analysis was done by R.L. and L.L. The paper was written by R.L., L.L., E.F.F., and C.A.E.

## Competing interests

The authors declare no competing interests.
