## [Peer Review File · Nature Communications]

Reviewers' Comments:

Reviewer #1:

Remarks to the Author:

This manuscript focuses on identifying mutations in a high-throughput format in transcription regulators that can enhance strain tolerance and strain productivity. The researchers first identified a global regulatory network comprised of regulatory genes that regulate or interact with hundreds or thousands of other genes in *E. coli*. In order to do this, the PIs employed a previously developed CRISPR-based method called CREATE that was developed previously in the Gill lab. By using CREATE, it was possible for them to construct a mutagenesis library targeting some of the regulators in the global network. From this data, the researchers were able to identify different characteristics important for industrial applications such as mutations that improved production phenotypes. To construct the network library, 82 regulators were selected to regulate or interact with approximately 4000 genes, specifically the active sites of these genes. This library was used to demonstrate large fold improvements, showing that the regulatory network library is both universally applicable and is an effective method for engineering targeted phenotypes.

Strengths:

This is a very well written manuscript. When looking at the overall flow, it is both easy to follow and succinct. This makes following the overall experimental goals and methodology simple.

In general, the experimental sections were well justified and easy to follow. For example, when designing the experiment to test the effect of alcohol-induced stress on the growth and survival rate of cells, the researchers explained their reasoning for choosing the mutant *FliA_R94K* as a way to improve cell stability under alcohol stress.

One of the strongest sections of the manuscript was the discussion. The discussion section was able to summarize and provide relevant analysis for all of the experiments conducted. Additionally, the discussion section was able to explain what was learned from each experiment as well as some applications for what was discovered.

The methods section was able to provide justification for why certain strains were used and protocols for growing all relevant cultures.

Figures 2-10 are easy to follow, and are full of relevant information. In particular, figure 6 is able to take a lot of information and display it in a clear, concise way. I was able to follow this graphic very easily. Additionally, figure 2 did a good job of explaining how genome editing efficiency is calculated and what it means in terms of the experiments.

Comments for improvement:

Line 46: The authors may want to include the following reference as an example of using transcription regulation to improve phenotypes and production: doi: 10.1186/s12934-016-0623-3.

While the grammar of the paper was mostly good, there were mistakes.

Line 127, therefore should either be at the start of the sentence or eliminated all-together.

Line 135 needs to be rewritten.

Line 150, there is a comma before the word "however" and it needs to be a semicolon.

Line 122: please explain how did you choose these concentrations. Would it be possible to use increasingly higher concentrations in order to identify the most critical and important mutations to improve tolerance and production?

Line 172 is a run on sentence

Line 298 "benefit" should be plural

The abstract states the paper will discuss the understanding of global regulation in less-understood organisms. This is brought up once at the very end of the paper and isn't really discussed how this library can be used in other organisms or even what these organisms could be. I think a small paragraph talking about this would make this claim more justified.

The introduction states tolerance trials were used but it does not explain why tolerance trials were used.

The section around line 303 is difficult to follow. There is a lot of information included there that may be easier to explain in a figure than written in the body of text.

The section "Mutations in OmpR improve isobutanol production" does not make clear why genes related to stress resistance being down-regulated would lead to the three mutations not being enriched in the tolerance selection.

Figure S1 looks interesting, but it is difficult to decipher any information other than there are many gene interactions between the regulators in the regulatory network. If this is all the figure is trying to show, it may make more sense to just have this as a sentence as following the lines in the figure is next to impossible.

In the Discussion section the authors may want to discuss the possibility of using the identified regulatory changes as a guide for further strain improvement.

Reviewer #2:

Remarks to the Author:

The authors present an impressive genetic engineering strategy for rewriting regulator networks. They use their previously described CREATE methodology to generate a designed library of 109,480 mutations of 82 regulators that control the expression for ~4,000 genes. The resultant strains were screened or selected for desired phenotypes - e.g., growth in furfural, styrene, acetate, isopropanol, and isobutanol and production of isobutanol.

I like the manuscript and think it is deserving of publication. I only have some minor suggestions for improvements. First, most of the figures have far too small font to read. Second, in figures like 1b, individual genes are used as examples of different classes. I think it would be better to just use a colored bar or list all genes of that type. Third, some discussion of what types of mutations were generated and enriched would be useful - i.e., how many enriched mutations were gain-of-function and how many were loss-of-function? Fourth, it would considerably strengthen the demonstration of this approach to show that mutations could be combined to either further improve a growth phenotype and/or metabolite production. Could the library be regenerated in the background of the high-performing first mutation? Similarly, or instead, would you pull out different mutants and/or still see improvements if you started with a pre-optimized strain from the phenotypes of interest? I wouldn't need to see this, but would view it as a nice demonstration of how this strategy can be serially used to achieve further improved phenotypes. Lastly, the description of "large fold increases" doesn't seem appropriate given that the improvement was only 1.4-fold in one case.

Dear editors and reviewers,

Thank you very much for your comments concerning our manuscript entitled “**Engineering regulatory networks for complex phenotypes in *E. coli***”. The comments were helpful for revising and improving our paper. We have studied the comments carefully and we made appropriate revisions. The main corrections are marked in red in this revised manuscript and our detailed responses to the comments are given below.

Reviewer #1 (Remarks to the Author):

This manuscript focuses on identifying mutations in a high-throughput format in transcription regulators that can enhance strain tolerance and strain productivity. The researchers first identified a global regulatory network comprised of regulatory genes that regulate or interact with hundreds or thousands of other genes in *E. coli*. In order to do this, the PIs employed a previously developed CRISPR-based method called CREATE that was developed previously in the Gill lab. By using CREATE, it was possible for them to construct a mutagenesis library targeting some of the regulators in the global network. From this data, the researchers were able to identify different characteristics important for industrial applications such as mutations that improved production phenotypes. To construct the network library, 82 regulators were selected to regulate or interact with approximately 4000 genes, specifically the active sites of these genes. This library was used to demonstrate large fold improvements, showing that the regulatory network library is both universally applicable and is an effective method for engineering targeted phenotypes.

Comments for improvement:

Line 46: The authors may want to include the following reference as an example of using transcription regulation to improve phenotypes and production: doi: 10.1186/s12934-016-0623-3.

Response: This reference 8 was added on page 2 line 45.

While the grammar of the paper was mostly good, there were mistakes.

Line 127, therefore should either be at the start of the sentence or eliminated all-together.

Response: We edited this sentence on page 5 line 128.

Line 135 needs to be rewritten.

Response: This sentence was re-written to clarify on page 5 line 137-138.

Line 150, there is a comma before the word “however” and it needs to be a semicolon.

Response: The comma was changed to a semi-colon on page 5 line 152.

Line 122: please explain how did you choose these concentrations. Would it be possible to use increasingly higher concentrations in order to identify the most critical and important mutations to improve tolerance and production? We performed growth selections with each sub-library under 2 g/L furfural, 300 mg/L styrene, 30 g/L acetate, 30 g/L isopropanol, and 8 g/L isobutanol because the cell growth clearly decreased at selected concentration based on the pre-test and other studies.

Response: We have modified the sentence on page 5 line 123-124.

Line 172 is a run on sentence

Response: We amended this sentence on page 6 line 175-176.

Line 298 “benefit” should be plural

Response: This grammatical error was corrected on page 10 line 296.

The abstract states the paper will discuss the understanding of global regulation in less-understood organisms. This is brought up once at the very end of the paper and isn't really discussed how this library can be used in other organisms or even what these organisms could be. I think a small paragraph talking about this would make this claim more justified.

Response: We added a short paragraph in the discussion (page 15 line 439-447).

The introduction states tolerance trials were used but it does not explain why tolerance trials were used.

Response: Tolerance trails were used because low tolerance is one of the limiting factors for improving productivity. In addition, screening methods are available for only a few traits of interest. Thus, tolerance selections with screening of positive variants for enhanced productivity is another option for improving productivity. We have modified the 3rd paragraph in the introduction section (page 3 line 61-66).

The section around line 303 is difficult to follow. There is a lot of information included there that may be easier to explain in a figure than written in the body of text.

Response: We added figure S14 to explain it better.

The section “Mutations in OmpR improve isobutanol production” does not make clear why genes related to stress resistance being down-regulated would lead to the three mutations not being enriched in the tolerance selection.

Response: The genes related to stress resistance in the alcohol tolerant strain were up-regulated, which might improve the alcohol tolerance. However, the genes related to stress resistance in OmpR strains were down-regulated. This could be the main reason why these strains were not enriched in the tolerance selection. We have added the related discussion on page 12 line 353-356.

Figure S1 looks interesting, but it is difficult to decipher any information other than there are many gene interactions between the regulators in the regulatory network. If this is all the figure is trying to show, it may make more sense to just have this as a sentence as following the lines in the figure is next to impossible.

Response: We have removed the Figure S1, and described it in the results section (page 3 line 87-88).

In the Discussion section the authors may want to discuss the possibility of using the identified regulatory changes as a guide for further strain improvement.

Response: We added this to the discussion (page 14 line 425-432).

Reviewer #2 (Remarks to the Author):

The authors present an impressive genetic engineering strategy for rewriting regulator networks. They use their previously described CREATE methodology to generate a designed library of 109,480 mutations of 82 regulators that control the expression for ~4,000 genes. The resultant strains were screened or selected for desired phenotypes - e.g., growth in furfural, styrene, acetate, isopropanol, and isobutanol and production of isobutanol.

I like the manuscript and think it is deserving of publication. I only have some minor suggestions for improvements.

First, most of the figures have far too small font to read.

Response: We have tried to re-organize the figures and use a bigger font for them.

Second, in figures like 1b, individual genes are used as examples of different classes. I think it would be better to just use a colored bar or list all genes of that type.

Response: The Circos plots in figures like 1b show each selection condition, and four highly enriched genes in each selection were labeled with different colors. They are not representative of classes. We apologize for this confusion and have clarified this in the figure legend.

Third, some discussion of what types of mutations were generated and enriched would be useful - i.e., how many enriched mutations were gain-of-function and how many were loss-of-function?

Response: We used PROVEAN (Protein Variation Effect Analyzer) (<http://provean.jcvi.org/index.php>) to evaluate the enriched mutations of *rstA* gene in the isobutanol tolerance and production selections. We have added more discussion in this on page 10 line 300-307.

Fourth, it would considerably strengthen the demonstration of this approach to show that mutations could be combined to either further improve a growth phenotype and/or metabolite production. Could the library be regenerated in the background of the high-performing first mutation? Similarly, or instead, would you pull out different mutants and/or still see improvements if you started with a pre-optimized strain from the phenotypes of interest? I

wouldn't need to see this, but would view it as a nice demonstration of how this strategy can be serially used to achieve further improved phenotypes.

Response: We agree with the reviewer that this would be a nice demonstration. Although we feel that further optimization of a specific phenotype is slightly outside the scope of this paper, we have added discussion about the possibility of making combinatorial mutations (page 14 line 425-432). As part of this discussion, we have also added a reference (from our lab) showing that the CREATE method used here can be done iteratively to further increase metabolite production (Liu, Liang, et al., 2018).

Lastly, the description of “large fold increases” doesn’t seem appropriate given that the improvement was only 1.4-fold in one case.

Response: We have removed the phrase “large fold increases” on page 13 line 396.